# Sniffing Threatening Open-World Objects in Autonomous Driving by Open-Vocabulary Models

### Yulin He
National University of Defense Technology
Changsha, China
heyulin@nudt.edu.cn

### Siqi Wang*
National University of Defense Technology
Changsha, China
wangsiqi10c@nudt.edu.cn

### Wei Chen*
National University of Defense Technology
Changsha, China
chenwei@nudt.edu.cn

### Tianci Xun
National University of Defense Technology
Changsha, China
xuntianci22@nudt.edu.cn

### Yusong Tan
National University of Defense Technology
Changsha, China
ystan@nudt.edu.cn

## Abstract

Autonomous driving (AD) is a typical application that requires effectively exploiting multimedia information. For AD, it is critical to ensure safety by detecting unknown objects in an open world, driving the demand for open world object detection (OWOD). However, existing OWOD methods treat generic objects beyond known classes in the train set as unknown objects and prioritize recall in evaluation. This encourages excessive false positives and endangers safety of AD. To address this issue, we restrict the definition of unknown objects to threatening objects in AD, and introduce a new evaluation protocol, which is built upon a new metric named U-ARecall, to alleviate biased evaluation caused by neglecting false positives. Under the new evaluation protocol, we re-evaluate existing OWOD methods and discover that they typically perform poorly in AD. Then, we propose a novel OWOD paradigm for AD based on fine-tuning foundational open-vocabulary models (OVMs), as they can exploit rich linguistic and visual prior knowledge for OWOD. Following this new paradigm, we propose a brand-new OWOD solution, which effectively addresses two core challenges of fine-tuning OVMs via two novel techniques: 1) the maintenance of open-world generic knowledge by a dual-branch architecture; 2) the acquisition of scenario-specific knowledge by the visual-oriented contrastive learning scheme. Besides, a dual-branch prediction fusion module is proposed to avoid post-processing and hand-crafted heuristics. Extensive experiments show that our proposed method not only surpasses classic OWOD methods in unknown object detection by a large margin (~3× U-ARecall), but also notably outperforms OVMs without fine-tuning in known object detection (~ 20% K-mAP). Our codes are available at https://github.com/harrylin-hyl/AD-OWOD.

*Corresponding author.

## CCS Concepts

• **Computing methodologies** → **Scene understanding**; *Information extraction.*

## Keywords

Auto-driving, Open World Object Detection, Open-Vocabulary Model

**ACM Reference Format:**
Yulin He, Siqi Wang, Wei Chen, Tianci Xun, and Yusong Tan. 2024. Sniffing Threatening Open-World Objects in Autonomous Driving by Open-Vocabulary Models. In *Proceedings of the 32nd ACM International Conference on Multimedia (MM '24), October 28-November 1, 2024, Melbourne, VIC, Australia.* ACM, New York, NY, USA, 10 pages. https://doi.org/10.1145/3664647.3680583

## 1 Introduction

Autonomous driving (AD) is a typical application that demands effective exploitation of multimedia information such as visual and linguistic modalities [13, 23, 24]. For AD, ensuring driving safety stands as a fundamental requirement, with one critical aspect being the detection of potential unknown objects in an open world. Recently, open world object detection (OWOD) task has been proposed to address this problem, aiming to bridge the gap between classic object detection (OD) [2, 34, 43] and practical OD in the open world. OWOD methods are expected to detect unknown objects beyond the known classes in the train set, which poses a challenge to evaluating their performance due to the extreme difficulty of defining "unknown objects" and labeling them all. To handle this issue, existing OWOD methods [8, 16] adopt a compromised way by re-organizing the original dataset, where a subset of classes is treated as known and the remainder as unknown. Meanwhile, recent works [30, 44] utilize the mean average precision of known classes (K-mAP) and the recall of the unknown class (U-Recall) to evaluate the performance of detecting known and unknown objects, respectively. However, U-Recall encourages the generation of high-confidence predictions to cover as many unknown objects as possible, regardless of whether many of them are false alarms (see Fig. 2). As a result, existing methods cannot satisfactorily address OWOD in AD. To this end, this paper focuses on this challenge and makes three-level contributions:

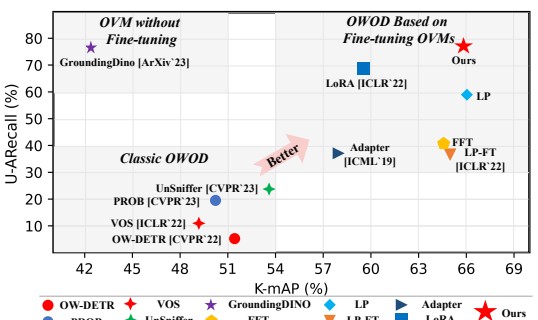

**Figure 1: Performance Comparison for AD-oriented OWOD. K-mAP and U-ARecall (details in Sec. 4) are used to evaluate the performance of known and unknown object detection, respectively. Classic OWOD methods exhibit a notable performance gap in U-ARecall. The OVM without fine-tuning excels in U-ARecall but falls short in K-mAP. Other OWOD methods based on fine-tuning OVMs improve K-mAP but show an apparent decline in U-ARecall. Compared to them all, our method achieves state-of-the-art overall performance in both U-ARecall and K-mAP.**

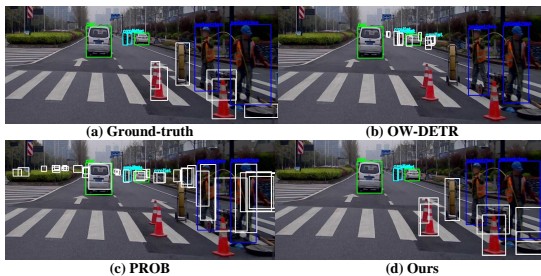

**Figure 2: Visualization comparison for AD-oriented OWOD. (a) Ground-truth, classic OWOD methods: (b) OW-DETR [8] and (c) PROB [44], along with (d) Our method. As can be seen, an abundance of false positives are notable in (b) and (c).**

(1) On the level of evaluation, we argue that ***unknown objects should be strictly restricted to those that likely emerge on the road and pose a threat to AD (i.e., threatening objects)***, such as roadblocks and wild animals, rather than all generic objects in previous OWOD works. By this definition, we are able to avoid introducing irrelevant objects and significantly reduce the difficulty of labeling. Meanwhile, we devise a new evaluation metric, termed average recall of the unknown class (***U-ARecall***) detailed in Sec. 4, to amplify the penalty for false positives including irrelevant objects to AD. This prevents the misguidance in original OWOD evaluation, which overemphasizes the recall of all unknown objects while neglecting the problem of false alarms due to incorrect recall. With the above definition and the proposed U-ARecall, we establish a new evaluation protocol for AD-oriented OWOD. Based on this evaluation protocol, we re-evaluate existing OWOD methods (see Fig. 1), and unveil that they typically yield low U-ARecall values, indicating their poor ability to accurately detect unknown objects.

(2) On the level of overall detection paradigm, we propose a novel OWOD paradigm based on ***fine-tuning foundational open-vocabulary models (OVMs)***, *e.g.*, GroundingDINO [28]. OVMs possess the capability to comprehend high-level language semantics

and recognize diverse visual patterns, which evidently contributes to detecting threatening objects in the open world. To further assist OVMs in comprehending the semantics of "threatening objects", we propose to concretize the concept of "threatening objects" by constructing a general vocabulary bag (see Sec. 5.1.2). This vocabulary bag is generally applicable to the AD scenario and contains vocabularies of possible threatening objects. Notably, we avoid using the unknown class vocabularies from the test set during the generation process, thus ensuring the generality of vocabulary bag and preventing information leakage. Meanwhile, we notice that OVMs tend to be "generalists" in generic scenarios rather than "specialists" in a specific scenario, which is reflected by the poor K-mAP of GroundingDINO in Fig. 1. Thus, OVMs need to be further fine-tuned in our paradigm to adapt the AD scenario.

(3) On the level of specific solution, we first identify two core challenges when fine-tuning OVMs for AD-oriented OWOD: 1) ***The risk of forgetting open-world generic knowledge***, *e.g.*, the OVM's performance in U-ARecall is degraded by 36% and 7.5% after being fine-tuned by fully fine-tuning (FFT) and LoRA [12] (see Fig. 1). 2) ***The effectiveness of acquiring scenario-specific knowledge***, *i.e.*, how to improve the K-mAP of OVMs while preserving a satisfactory U-ARecall performance. To address the two challenges above, we propose a brand-new solution that includes the following novel techniques: 1) A dual-branch architecture (DBA) that contains a frozen and a trainable branch to preserve general open-world knowledge by freezing the parameters of OVMs. 2) A visual-oriented contrastive learning scheme to enhance the effectiveness of acquiring scenario-specific knowledge by minimizing the contrastive loss between training samples and high-confidence visual exemplars. Besides, we propose a prediction fusion module to integrate predictions of two branches in DBA by aligning their query positions, thereby avoiding hand-crafted post-processing.

In summary, our contributions are four-fold: (1) We devise a more suitable evaluation protocol for AD-oriented OWOD, which includes restricting unknown objects to threatening objects in AD and introducing a new evaluation metric to alleviate the biased evaluation in OWOD. Based on this evaluation protocol, we re-evaluate existing OWOD methods and establish the AD-oriented OWOD benchmark. (2) We propose a new OWOD paradigm for AD based on fine-tuning OVMs, exploiting rich prior knowledge, including high-level language semantic and diverse visual patterns, to distinguish between threatening objects and false positives. (3) We identify and address two core challenges when fine-tuning OVMs: preserving open-world generic knowledge by dual-branch architecture and acquiring scenario-specific knowledge by visual-oriented contrastive learning. (4) We propose a prediction fusion module that can integrate predictions from multiple branches without the need of post-processing and hand-crafted heuristics, serving as a generalized method applicable to transformer-based detectors.

## 2 Related Works

**Open World Object Detection**. Open world object detection (OWOD) aims to adapt classic object detection to the open-world setting. In this setting, methods are required to detect unknown objects beyond the known classes in the train set. OWOD task initially proposed by Joseph *et al.* [16] has garnered significant

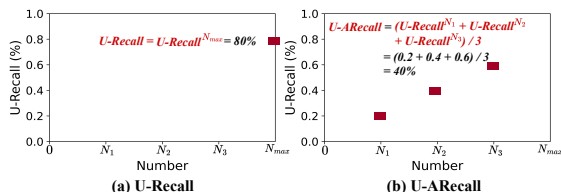

**Figure 3: Examples of calculation process for (a) U-Recall and (b) the proposed U-ARecall. For simplicity, we sample three values of $N$ in U-ARecall for demonstration.**

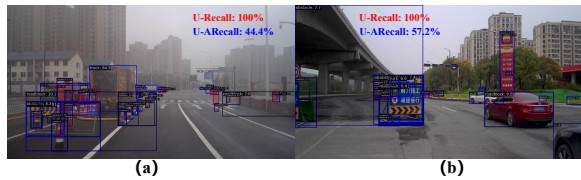

**Figure 4: Examples to show the differences in value between U-Recall and U-ARecall for the same predictions.**

attention [8, 30, 31, 36, 39, 44]. ORE [16] adapted Faster-RCNN [34] detector with an energy based unknown identifier for the OWOD objective. Subsequently, OW-DETR [8] adapted Transformer-based detector (D-DETR [43]) into OWOD, employing a pseudo-labeling scheme to guide the detection of unknown objects. PROB [44] then introduced a probabilistic objectness head to identify unknown objects by estimating their likelihood. VOS [7] modeled a decision boundary to distinguish between unknown objects by adaptively synthesizing virtual outliers. UnSniffer [25] established a generalized object confidence score to separate non-object and object classes. Despite these efforts, existing classic OWOD methods tend to yield an intolerably high false alarm rate for AD. Meanwhile, OWOD research for the AD scenario is also underway. ORDER [36] is a pioneering work in extending OWOD to AD, introducing a feature-mix method to enhance the semantics of unknown objects. However, it followed the original evaluation in classic OWOD, and its code is not available. SalienDet [6] utilized the saliency map to improve the capabilities of unknown object detection, but the saliency map is not always available in practical application. In [31], they explored the zero-shot object detection in AD by exploiting the language model BERT [5], but did not consider detecting open-world unknown objects. Besides, its code is also not available. Therefore, there is an urgent demand for a comprehensive AD-oriented OWOD benchmark to advance the research of this field.

**Open-vocabulary Model**. Open-vocabulary model (OVM) aims to expand the recognizable classes of models by constructing a open vocabulary bag. Recently, vision-language pretraining [14, 33] has become a popular paradigm for OVMs to learn rich prior knowledge from large amounts of raw image-text pairs. CLIP [33] is a representative OVM work, which has shown strong zero-shot recognition ability in open vocabulary classification (OVC). In addition to OVC, OVMs have also been adapted into various tasks, *e.g.*, open vocabulary detection (OVD) [21, 28]. In OVD, OVR-CNN [41] first adopted BERT [5] to pre-train the detector on image-caption pairs and then fine-tuned the model for the downstream detection task. Following it, significant efforts [18, 35, 38] have been made to enhance the OVD benchmark. Visual grounding is another related research direction, first proposed by GLIP [21], which reformulated object

detection as a phrase grounding problem. GroundingDINO then extended the advanced transformer-based detector DINO [42] to OVD by performing vision-language modality fusion. While OVMs have achieved high accuracy on generic scenarios [19, 27], they are more like "generalists" rather than "specialists", and perform poorly in some specific scenarios. Therefore, it is valuable to fine-tune OVMs to adapt the specific scenarios, *e.g.*, autonomous driving.

**Parameter-efficient Fine-tuning**. Fine-tuning is a common strategy to adapt models to specific scenarios. Fully fine-tuning (FFT) is widely used to make the entire network trainable, but it is time-consuming and prone to overfitting when trained on small datasets. Efficient transfer leaning (ETL) attempts to address this issue by updating or adding a small set of trainable parameters, limiting the dimension of the optimization problem to prevent catastrophic forgetting [32]. For methods without introducing new parameters, linear probing (LP) learned a linear probe on top of frozen embeddings. LP-FT [17] further adopted a two-stage fine-tuning, first performing LP and then FFT with the weight of classifier initialized in the first stage. Alternatively, some methods introduced new parameters within the network [11, 12, 15, 22]. Adapter [11], as an addition-based method, incorporated a bottleneck adapter structure into the transformer blocks. While LoRA [12], as a reparameterization-based method, optimized two low-rank matrices, and then merged them into the weight matrices. Currently, most ETL methods are centered around natural language processing (NLP), with relatively limited research on OVMs.

## 3 Preliminaries of OWOD

Referring to [16, 25], the problem of OWOD is formulated as follows: a model is trained by a dataset with ***known objects*** from a set of known classes $C_k = \{1, 2, ..., C\}$. During inference, the model is required to detect both known objects from $C_k$ and ***unknown objects***, which are classified into an unknown class $C_u = \{C + 1\}$. This formulation also applies to the AD scenario in this paper.

Theoretically, "unknown objects" is a vague concept that refers to all generic objects outside known classes $C_k$, so it is extremely difficult to precisely define unknown objects and label them all. To bypass this difficulty, previous methods typically perform OWOD by re-organizing existing multi-class object detection datasets (*e.g.*, Pascal VOC [10] and MS COCO [27]), *i.e.*, treating a subset of classes as known and the remainder as unknown, which avoids defining and labeling unknown objects by themselves. However, such re-organization does not overcome the difficulty of defining unknown objects from the root, and many of detected unknown objects are irrelevant to the specific application scenario like AD.

As to the evaluation of OWOD, recent works like [8, 30, 44] tend to adopt mAP of known classes ***(K-mAP)*** and recall of the unknown class ***(U-Recall)*** to assess the performance of known and unknown object detection, respectively. Specifically, AP is equal to the area under precision-recall (PR) curve for one class, and K-mAP denotes the mean of AP across all known classes. As to the detection of unknown objects, its U-Recall is yielded by computing the recall with 100 predictions, *i.e.*, U-Recall$^{100}$. However, U-Recall$^{100}$ does not penalize false positives within predictions. Since 100 is often much larger than the number of objects in most cases, existing OWOD methods tend to output as many high-confidence predictions as

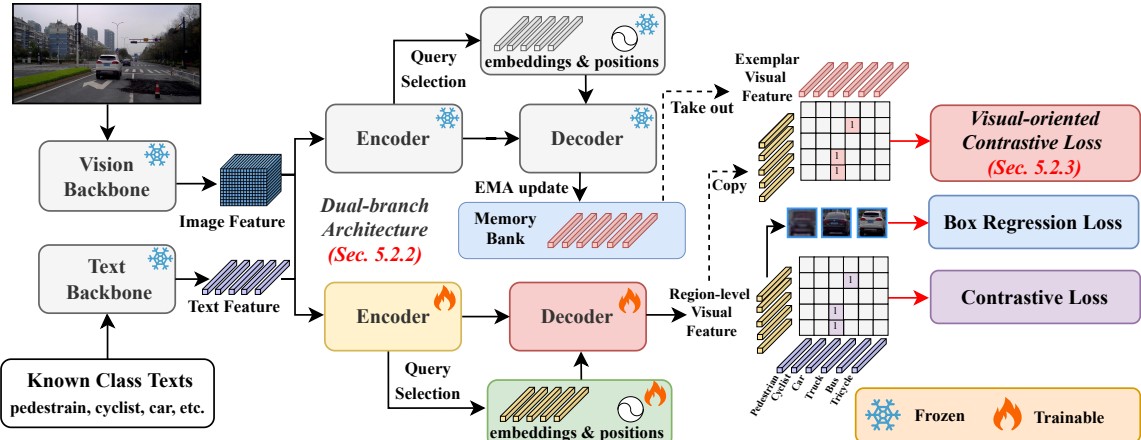

**Figure 5: The training architecture of our method. It includes a vision and a text backbone for extracting image and text features, an encoder for fusing multi-modality features, a decoder for generating region-level visual features, and a head network for classifying and regressing detection predictions. The only trainable component is a branch (marked by "fire") of dual-branch architecture. The visual-oriented contrastive loss is proposed to enhance the effectiveness of fine-tuning OVMs.**

possible to cover more unknown objects, regardless of whether many of them are false positives (see Fig. 2). This will incur higher false alarm rate and pose a grave threat to the safety of AD.

## 4 The Proposed Evaluation Protocol

To enable a properer evaluation of OWOD in AD, we propose a new evaluation protocol, which mainly includes redefining unknown objects and designing a more valid evaluation metric. Based on this new evaluation protocol, we re-evaluate existing OWOD methods and establish the AD-oriented OWOD benchmark.

To tackle the vagueness of "unknown objects", we propose to strictly restrict them to ***threatening objects***, *i.e.*, objects that likely emerge on the road and pose a threat to AD. By this definition, we can mitigate the ambiguity when labeling unknown objects for OWOD in AD, as the concept of "threatening objects" is more definite than that of "unknown objects". Besides, we can also ensure that the detected unknown objects are tightly relevant to the AD scenario. In this way, our definition enables more meaningful OWOD for AD than previous research like [8, 16, 30, 36, 44].

Having redefined "unknown objects" for OWOD in AD, we intend to devise a new evaluation metric to amplify the penalty for false positives (*e.g.*, distracting background and irrelevant objects to AD). As stated in Sec. 3, we observe that the number of predictions used in evaluation ($N$) usually poses a significant impact on the value of recall, *i.e.*, allowing more predictions for evaluation often leads to a higher recall, but it may also introduce more false positives. Previous methods typically use U-Recall[100] that simply sets $N$ to a relatively large number (100), which overemphasizes the recall but ignores false positives. To address this problem, given the maximum number of unknown objects $N_{max}$ in the test set ($N_{max}$ is available in evaluation), we first sample $r$ values of $N$ from 0 to $N_{max}$: $0 < N_1 < ... < N_r < N_{max}$ at an equal interval. Then, we propose a new evaluation metric named ***U-ARecall*** to evaluate the performance of unknown object detection:

$$\text{U-ARecall} = \frac{\sum_{i=1}^{r} \text{U-Recall}^{N_i}}{r}, \tag{1}$$

where U-Recall$^{N_i}$ is the recall value with $N_i$ predictions.

By virtue of U-ARecall, when only a small number of predictions are allowed for evaluation, we can penalize those high-confidence false positives by a low recall in this case. Meanwhile, we are also able to yield a more comprehensive assessment of recall by varying the value of $N$, so as to handle images with different number of unknown objects. In this way, U-ARecall amplifies the penalty for excessive false positives, thereby providing a more reasonable measurement for OWOD in AD compared to frequently-used U-Recall[100], simplified as U-Recall (see Fig. 4). In addition, we introduce a metric named ***UK-Mean*** to assess the overall performance of both known and unknown object detection:

$$\text{UK-Mean} = \beta \cdot \text{K-mAP} + (1 - \beta) \cdot \text{U-ARecall}, \tag{2}$$

where $\beta$ is a parameter to trade off the importance of K-mAP and U-ARecall. In our setup, $\beta$ is set to 0.5. We also carefully select appropriate data sets from the AD scenario (details in Sec. 6.1), which are combined with the aforementioned redefinition and new metric to construct a new benchmark for AD-oriented OWOD.

## 5 Methodology

### 5.1 The Proposed OWOD Paradigm

*5.1.1 Motivation.* With our new benchmark above, we extensively re-evaluate previous OWOD methods for AD-oriented OWOD. Unfortunately, those methods typically perform unsatisfactorily (see Fig. 1). The reason is that the detection paradigms of all existing OWOD methods fail to exploit the knowledge from the open world, which is exactly the key to discovering open-world unknown objects. To address this issue, we propose a new OWOD paradigm based on fine-tuning OVMs, which possess the knowledge to comprehend rich high-level language semantics and recognize diverse visual patterns in the open world. Our OWOD paradigm consists of two stages: 1) generating a vocabulary bag to determine the range of detection classes. 2) fine-tuning OVMs to adapt the AD scenario.

*5.1.2 Vocabulary Bag Generation.* To ensure generality and simplicity, we generate a general vocabulary bag to concretize the

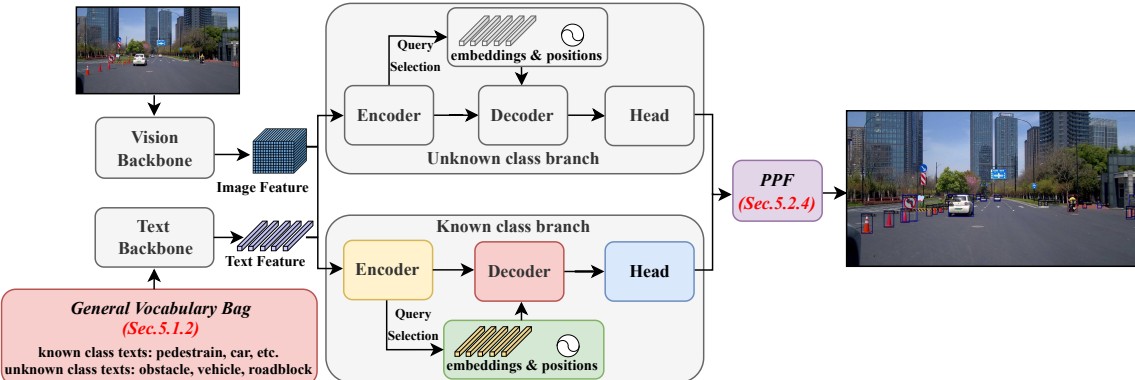

**Figure 6: The Inference architecture of our method. The general vocabulary bag, including known and unknown class texts, is used to generate text features. Meanwhile, the input image is used to produce image features. With these features above, the known and unknown branches generate respective predictions. Finally, the PPF module fuses the predictions of two branches.**

semantics of "unknown objects", so as to correctly distinguish between unknown objects and false positives. Specifically, we define three coarse classes according to the common sense in the traffic scenario, *i.e.*, {"*obstacle*", "*vehicle*", "*roadblock*"}. Despite the simplicity, those coarse classes have already covered most unknown objects on the road, as they are relatively abstract and each of them semantically includes many sub-classes. Meanwhile, our vocabulary bag avoids using any unknown class text from the test set, thereby avoiding information leakage. Therefore, the vocabulary bag is highly generalizable and not specific to a certain dataset. Pleasantly surprising, this general vocabulary bag is simple yet effective, which exhibits excellent performance in our later experiments.

*5.1.3 Fine-tuning OVMs.* By pre-training on massive image-text data, OVMs enjoy a strong ability to recognize diverse objects in the open world. However, OVMs tend to be "generalists" rather than "specialists". In the specific scenario like AD, OVMs suffer from a notable performance gap in detecting known objects (see K-mAP in Fig. 1). Thus, our OWOD paradigm fine-tunes OVMs to adapt the current AD scenario. We select the recently proposed GroundingDINO [28] as the default OVM, as it achieves state-of-the-art performance across various zero-shot OD benchmarks [19, 27].

## 5.2 The Proposed OWOD Solution

*5.2.1 Motivation.* Following the proposed OWOD paradigm, we customize some classic fine-tuning techniques like linear probe (LP), as well as some recent fine-tuning methods like LoRA [12] and Adapter [11], specifically for AD-orientated OWOD. However, as shown in Fig. 1, they either suffer from catastrophically forgetting open-world generic knowledge (*e.g.* LP-FT), or fail to effectively learn scenario-specific knowledge (*e.g.*, LoRA). Therefore, we identify those challenges and tackle them by corresponding techniques: 1) the maintenance of open-world generic knowledge by a dual-branch architecture (DBA) in Sec. 5.2.2. 2) the acquisition of scenario-specific knowledge by a visual-oriented contrastive learning (VorCL) scheme in Sec. 5.2.3. Besides, a post-processing-free prediction fusion (PPF) module is introduced in Sec. 5.2.4 to integrate the dual-branch predictions. Based on those novel techniques, we propose a brand-new OWOD solution, the training and inference procedures of which are elucidated in Fig. 5 and Fig. 6.

*5.2.2 Dual-branch Architecture.* To mitigate catastrophic forgetting, the proposed DBA first uses the frozen vision/text backbone to extract image/text features, which are then fed into a dual-branch architecture that consists of a frozen unknown class branch and a trainable known class branch. With the frozen backbones and the frozen unknown class branch, DBA can effectively preserve the generic knowledge within OVMs to handle the unknown objects in the open world. Meanwhile, the trainable known class branch enables DBA to acquire the specific knowledge of the AD scenario for detecting known objects. Since the number of parameters in the known class branch is much less than the OVM, the computational cost of DBA is affordable. Besides, DBA is parallelizable and can be optimized to reduce inference time during deployment.

Let us define $I$ as the input image, $T$ as the input text, $\mathcal{B}_v$ as the vision backbone, $\mathcal{B}_t$ as the text backbone. Since the components of both the known class branch and the unknown class branch are identical, we use the same notation for their corresponding components for simplicity. Let $\mathcal{E}$ and $\mathcal{D}$ be the encoder and decoder of the branch. $\mathcal{F}$ denotes a query selection function that selects top-k confident query embeddings from the output of $\mathcal{E}$, and $\mathbf{p}$ is the query positions. The forward pass of a branch starts by:

$$\mathbf{F}, \mathbf{r}^t = \mathcal{E}(\mathcal{B}_v(I), (\mathcal{B}_t(T)) \tag{3}$$

$$\mathbf{r}^b = \mathcal{D}(\mathbf{F}, \mathcal{F}(\mathbf{F}), \mathbf{p}) \tag{4}$$

where $\mathbf{F}$ and $\mathbf{r}^t$ are the enhanced image and text features, respectively. $\mathbf{r}^b$ is the region-level visual features at positions $\mathbf{p}$. Then, the detection scores are calculated by the classification head $\mathcal{H}^{cls}$:

$$\mathbf{s} = \mathcal{H}^{cls}(\mathbf{r}^t, \mathbf{r}^b) = Sigmoid(T \cdot \langle \mathbf{r}^t \cdot \mathbf{r}^b \rangle), \tag{5}$$

where $\langle \cdot \rangle$ is the similarity function that computes the inner product. $T$ is the temperature to re-scale the value, set to 1 as [28]. Bounding boxes can be obtained by the regression head $\mathcal{H}^{reg}$: $\mathbf{b} = \mathcal{H}^{reg}(\mathbf{r}^b)$.

*5.2.3 Visual-oriented Contrastive Learning.* We begin by reviewing the standard contrastive learning [33] in OVMs, which minimizes the distance between visual features and textual features. It uses the detection scores $\mathbf{s}$ in Eq. 5 to calculate the contrastive loss. For example, the Focal loss [26] for contrastive learning is:

$$\mathcal{L}_{cl} = -y(1-\mathbf{s})^\gamma \log(\mathbf{s}) - (1-y)\mathbf{s}^\gamma \log(1-\mathbf{s}), \tag{6}$$

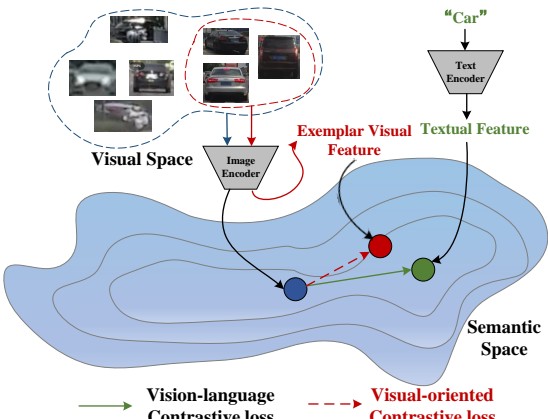

**Figure 7: Illustration of the basic idea behind visual-oriented contrastive learning. It generates exemplar visual features from high-confidence samples, and treats them as visual semantic centers to optimize the other hard samples.**

where $y$ denotes the labels of known classes, and $\gamma$ is a hyper-parameter that is set to 2 [28].

To further enhance the effectiveness of fine-tuning via contrastive learning, we propose the VorCL scheme that aims to minimize the distance between visual features from training samples and high-confidence exemplar visual features. These exemplar visual features serve as semantic centers to optimize hard samples, as illustrated in Fig. 7. The known class branch is the only trainable part of our solution, while its enhanced text features $\mathbf{r}^t$, region-level visual features $\mathbf{r}^b$ and corresponding scores $\mathbf{s}$, can be obtained by Eq. 3, Eq. 4 and Eq. 5, respectively. Let $N_q$ be the number of queries per image and $B$ be the training batch size. The $c$-th class exemplar visual feature $(\mathbf{r}^v)_c$ can be obtained as follows:

$$(\mathbf{r}^v)_c = \frac{\sum_{i=1}^{N_q B} \mathbb{1}(\max((\mathbf{s}_i)_c) \geq \tau)(\mathbf{r}_i^b)_c \cdot (\mathbf{s}_i)_c}{\sum_{i=1}^{N_q B} \mathbb{1}(\max((\mathbf{s}_i)_c) \geq \tau)(\mathbf{s}_i)_c} \quad (7)$$

where $(\mathbf{s}_i)_c$ and $(\mathbf{r}_i^b)_c$ are $\mathbf{s}$ and $\mathbf{r}^b$ for $i$-th query and $c$-th class, respectively. $\tau$ is a selection threshold used to select $\mathbf{r}^v$. As described in Eq. 7, the $\mathbf{r}^b$ with the same class will be weighted by their corresponding scores $\mathbf{s}$ and summed, producing $\mathbf{r}^v$ in the current training batch. Then, we adopt a memory bank with the length of known classes to store and update $\mathbf{r}^v$. The memory bank is initialized by $\mathbf{r}^t$ and continuously updated with the input training batch. In the next training batch, the $(\mathbf{r}^v)_c$ at the $t$-th iteration is updated by exponential moving average (EMA):

$$(\mathbf{r}_t^v)_c = (1 - \lambda)(\mathbf{r}_t^v)_c + \lambda(\mathbf{r}_{t-1}^v)_c, \quad (8)$$

where $\lambda$ is an EMA hyper-parameter. Then, the visual-oriented detection scores $\mathbf{s}_v$ are computed by: $\mathbf{s}_v = \mathcal{H}^{cls}(\mathbf{r}^v, \mathbf{r}^b)$. The calculation process of VorCL loss is analogous to Eq. 6 but uses $\mathbf{s}_v$:

$$\mathcal{L}_{vorcl} = -y(1 - \mathbf{s}_v)^\gamma \log(\mathbf{s}_v) - (1 - y)\mathbf{s}_v^\gamma \log(1 - \mathbf{s}_v). \quad (9)$$

Based on Eq. 6 and Eq. 9, the total loss of our OWOD solution is:

$$\mathcal{L}_{total} = \mathcal{L}_{reg} + \mathcal{L}_{cl} + \mathcal{L}_{vorcl} \quad (10)$$

where $\mathcal{L}_{reg}$ is the regression loss as in [28].

*5.2.4 Post-processing-free Prediction Fusion.* PPF module aims to integrate the predictions of the known class branch and the unknown class branch to produce the final predictions. In general, non-maximum suppression (NMS) is a prevalent module to fuse two distinct sets of predictions in OD. However, it introduces hand-crafted heuristics, which contradicts the core idea of transformer-based detectors (GroundingDINO also falls to this category), *i.e.*, minimizing the manual design. This motivates us to propose a new fusion technique without the need of hand-crafted post-processing.

By inspecting the positional variation of proposals and bounding boxes, we found that the misalignment between predictions of the known and the unknown branch arises from the differences in query positions, *i.e.*, $\mathbf{p}$ in Eq. 4. Query positions represent the positions of learnable proposals, which are continuously updated and changed during training, thereby leading this misalignment problem. By applying the same query position for both branches, their predictions can be well aligned as a one-to-one pairing, so as to achieve one-to-one fusion. Specifically, we use the query positions of the unknown branch $\mathbf{p}_u$ for both branches. This is because $\mathbf{p}_u$ is class-agnostic and can already cover the majority of objects. As mentioned in Sec. 5.2.2, we can obtain the detection scores and bounding boxes of the known branch $(\mathbf{s}_k, \mathbf{b}_k)$ and the unknown branch $(\mathbf{s}_u, \mathbf{b}_u)$. Then, we apply the geometric mean to combine $\mathbf{s}_k$ and $\mathbf{s}_u$. The $c$-th class final detection scores $(\mathbf{s}_f)_c$ are calculated by:

$$(\mathbf{s}_f)_c = \begin{cases} (\mathbf{s}_k)_c^{(1-\alpha)} \cdot (\mathbf{s}_u)_c^{\alpha} & \text{if } c \in C_k \\ (\mathbf{s}_u)_c & \text{if } c \in C_u \end{cases} \quad (11)$$

where $\alpha$ is the hyper-parameter to control the contributions of $(s_k)_c$ and $(s_u)_c$. Subsequently, we adopt a class-wise box selection operation to fuse $\mathbf{b}_k$ and $\mathbf{b}_u$. The $c$-th class final bounding boxes $(\mathbf{b}_f)_c$ are calculated as follows:

$$(\mathbf{b}_f)_c = \begin{cases} (\mathbf{b}_k)_c & \text{if } c \in C_k \\ (\mathbf{b}_{uk})_c & \text{if } c \in C_{uk} \end{cases} \quad (12)$$

## 6 Experiment

### 6.1 Datasets

For the evaluation of AD-oriented OWOD, we survey existing AD datasets [1, 4, 9, 20, 37, 40] and discover that, to our best knowledge, only the labeling criteria of the CODA dataset [20] align well with our definition of threatening objects. However, CODA is originally designed for corner case detection, and corner cases in validation set are accessible, which contradicts to the problem setting of OWOD that open-world objects are strictly unknown during training. Therefore, in our evaluation protocol, we treat the common classes of CODA as known classes and its corner-case classes as unknown classes. We leverage the released validation set of CODA as the test set of our evaluation protocol. It consists of 4,484 road driving images with 29 representative classes, including 6 common classes and 23 corner-case classes.

We fine-tune OVMs by the train set of the SODA [9] or the BDD100K [40] dataset. The two datasets are used to simulate the case with small or large domain differences from CODA. In this way, we can inspect the generalization abilities of methods. Specifically, SODA includes 10,000 images with 6 classes, while BDD100K contains 70,000 images with 10 classes. All classes in the train set of

**Table 1: Performance comparison using SODA for the AD-oriented OWOD benchmark. U-R means U-Recall. FR and R-OVM represent FasterRCNN and Raw OVM, respectively.**

| Method | U-R[10] | U-R[20] | U-R[30] | U-ARecall | K-mAP | UK-Mean |
|---|---|---|---|---|---|---|
| FR[34] | 0. | 0. | 0. | 0. | 53.4 | 26.7 |
| D-DETR[43] | 0. | 0. | 0. | 0. | 53.3 | 26.6 |
| PROB[44] | 12.9 | 20.1 | 25.0 | 19.3 | 50.2 | 34.7 |
| OW-DETR[8] | 2.6 | 5.4 | 8.3 | 5.4 | 51.4 | 28.4 |
| VOS[7] | 8.9 | 11.4 | 13.0 | 11.1 | 49.2 | 30.1 |
| UnSniffer[25] | 20.6 | 24.7 | 25.4 | 23.6 | 53.7 | 38.6 |
| R-OVM[28] | 67.8 | 78.7 | 83.0 | 76.5 | 42.4 | 59.4 |
| FFT | 33.3 | 41.7 | 46.6 | 40.5 | 64.6 | 52.5 |
| LP | 50.6 | 61.2 | 65.2 | 59.0 | 66.1 | 62.5 |
| LP-FT[17] | 30.1 | 38.4 | 44.1 | 37.5 | 65.0 | 51.2 |
| Adapter[11] | 31.3 | 38.2 | 42.4 | 37.3 | 57.9 | 47.6 |
| LoRA[12] | 60.7 | 71.5 | 74.9 | 69.0 | 59.5 | 64.2 |
| **Ours** | 68.2 | 79.6 | 84.1 | **77.3** | 65.9 | **71.6** |

**Table 2: Performance comparison using BDD100K for the AD-oriented OWOD benchmark. U-R denotes U-Recall. R-OVM means Raw OVM. Only classic OD and OWOD methods with the best performance are included due to page limit.**

| Method | U-R[10] | U-R[20] | U-R[30] | U-ARecall | K-mAP | UK-Mean |
|---|---|---|---|---|---|---|
| FR[34] | 0. | 0. | 0. | 0. | 53.3 | 26.6 |
| UnSniffer[25] | 11.3 | 21.3 | 27.3 | 20.0 | 55.3 | 37.6 |
| R-OVM[28] | 55.2 | 71.5 | 78.8 | **68.5** | 42.2 | 55.3 |
| FFT | 34.5 | 40.4 | 43.6 | 39.5 | 60.3 | 49.9 |
| LP | 34.6 | 41.5 | 45.1 | 40.4 | 60.7 | 50.5 |
| LP-FT[17] | 30.0 | 37.0 | 39.9 | 35.6 | 60.6 | 48.1 |
| Adapter[11] | 37.0 | 44.7 | 48.8 | 43.5 | 60.1 | 51.8 |
| LoRA[12] | 44.9 | 61.5 | 68.8 | 58.4 | 56.4 | 57.4 |
| **Ours** | 54.9 | 71.3 | 78.8 | 68.3 | 60.5 | **64.4** |

SODA are viewed as known classes. We view 5 of the 10 classes of BDD100k as known classes (*i.e.*, person, rider, car, bus, and truck), as only those 5 classes appear in common classes of CODA.

## 6.2 Implementation Details

As to the setting of U-ARecall, we get the maximum value of labeled unknown objects in CODA ($N_{max}$ = 43), and adopt three sampling prediction numbers ($N_1$ = 10, $N_2$ = 20, $N_3$ = 30) to handle images with different numbers of unknown objects. Due to the superior open-world capabilities of GroundingDino [28], we choose it as the OVM for further fine-tuning, which is implemented by MMDetection [3]. We train the model over 12 epochs with the adamW optimizer [29], and its weight decay is set to $1 \times 10^{-4}$. The learning rate is initialized by $1 \times 10^{-4}$ and reduced by a factor of 10 after reaching the 11-th epoch. Without careful selection, we set the hyper-parameters in VorCL and PPF based on our intuition and experience: $\tau = 0.3$ in Eq.7, $\lambda = 0.99$ in Eq.8, and $\alpha = 0.2$ in Eq. 11.

## 6.3 Comparison with State-of-the-art Methods

In our AD-oriented OWOD benchmark, as shown in Tab. 1 and Tab. 2, methods under comparison are categorized into five types: classic OD methods (the first part), classic OWOD methods (the second part), the raw OVM [28] without fine-tuning (the third part), the OVM directly fine-tuned by standard fine-tuning methods (the

fourth part) and our method (the last part). We list results of classic OD methods to show their inability to handle OWOD, and results of classic OWOD methods to demonstrate the greater difficulty of AD-orientated OWOD than classic OWOD. As our method is based on OVM fine-tuning, we also include the raw OVM and OVM fine-tuned by standard methods to present a fairer comparison. Methods employed for fine-tuning the OVM include fully fine-tuning (FFT), linear probing (LP), LP-FT [17], Adapter [11], and LoRA [12].

The experimental results using SODA dataset are shown in Tab. 1, which reveal that our method achieves state-of-the-art performance for AD-oriented OWOD. As for the ***overall performance (UK-Mean)***, our method significantly improves UK-Mean by more than 33 % compared to the best classic OWOD method (UnSniffer), which justifies the necessity of exploiting the OVM's rich open-world knowledge. Even compared to the second-best method (LoRA), our method also notably surpasses it by 7.4% in UK-Mean, which indicates the effectiveness of our proposed techniques. When we look at the ***performance of unknown object detection (U-ARecall)*** alone, our method has achieved an remarkable performance advantage over classic OWOD methods, and it is even slightly better than the raw OVM by 0.8%. In particular, our method evidently outperforms the OVM fine-tuned by standard methods by at least 8.3% U-ARecall, which demonstrates that our method can effectively mitigate the problem of catastrophic forgetting during fine-tuning, thereby preserving the open-world generic knowledge within the OVM. As for the ***performance of known object detection (K-mAP)***, our method notably improves K-mAP by 23.5% when compared with the raw OVM, which unveils the necessity of fine-tuning OVMs to adapt the AD scenario. More importantly, our method performs comparably to OVM fine-tuned by standard methods ($\leq$ 0.2% K-mAP). Such results suggest that our method can be as effective as standard fine-tuning methods in acquiring scenario-specific knowledge, without being severely influenced by catastrophic forgetting like those methods. As a consequence, our method achieves fairly satisfactory performance in detecting both unknown and known objects. In addition, the experimental results using BDD100K (see Tab. 2) exhibit a very similar trend to that of SODA. This proves the effectiveness of our method again, and verifies the fine generalization ability of our method across different domain gaps.

Fig. 8 depicts the qualitative results of Raw OVM, FFT and our method. We set a threshold of 0.3 to determine the visualization of known and unknown objects. As can be seen, the raw OVM tends to overlook or misidentify certain known objects, *e.g.*, pedestrian, and bus. FFT typically fails to detect some unknown objects, *e.g.*, traffic cone, stroller, and dog. In contrast, our method exhibits a strong ability to accurately detect both known and unknown objects, demonstrating its robust open-world recognition capabilities.

## 6.4 Discussions

**Ablation Studies**. We analyze the contributions of our proposed DBA, VorCL and PPF, with the experimental results presented in Tab 3. When we apply the DBA, U-ARecall improves by a large margin (27.7%), which unveils its advantage in alleviating knowledge forgetting. Replacing NMS with PPF leads to notable improvements in both U-ARecall and K-mAP by 9.4% and 6.4%, respectively, demonstrating the strength of avoiding hand-crafted heuristics in NMS.

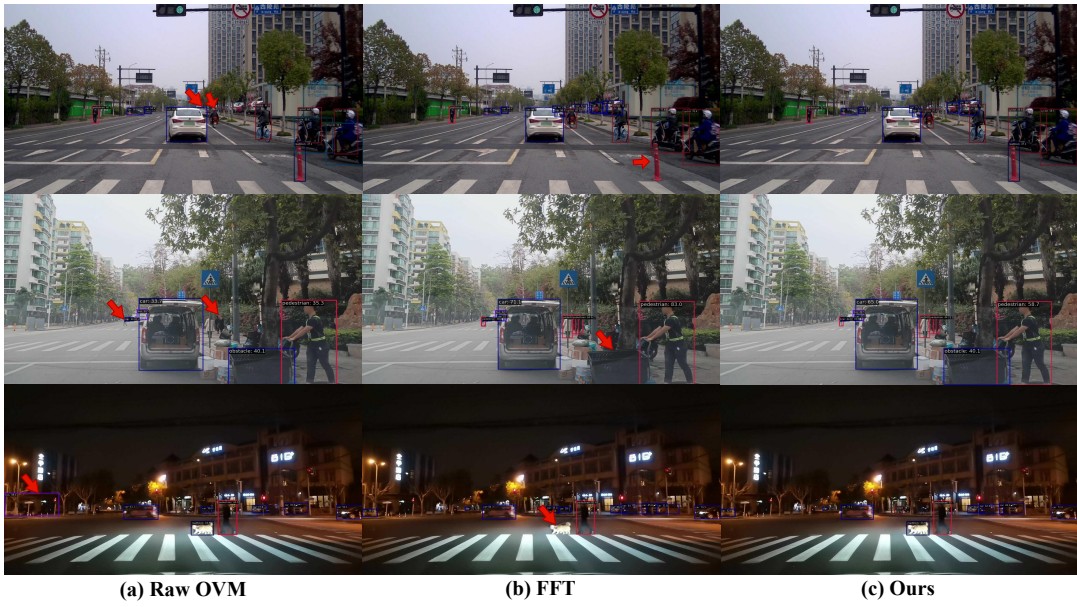

(a) Raw OVM                          (b) FFT                          (c) Ours

**Figure 8: Comparison of visualization results. Missing objects and prediction errors are marked by red arrows.**

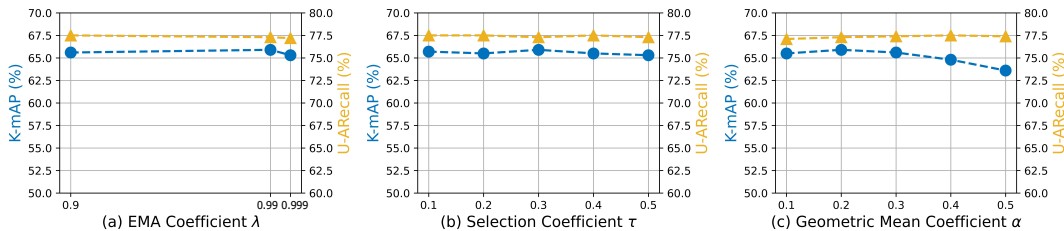

(a) EMA Coefficient $\lambda$          (b) Selection Coefficient $\tau$          (c) Geometric Mean Coefficient $\alpha$

**Figure 9: Sensitivity analysis of the hyper-parameters in (a) $\lambda$ and (b) $\tau$ for VorCL, along with (c) $\alpha$ for PPF.**

**Table 3: Ablation study of our proposed techniques. F. is the prediction fusion module, *e.g.,* NMS and PPF.**

| DBA | VorCL | F. | U-ARecall | K-mAP | UK-Mean |
|---|---|---|---|---|---|
| ✗ | ✗ | None | 40.5 | 64.6 | 52.5 |
| ✓ | ✗ | NMS | 68.2 | 58.4 | 63.3 |
| ✓ | ✗ | PPF | **77.6** | 64.8 | 71.2 |
| ✓ | ✓ | PPF | 77.3 | **65.9** | **71.6** |

**Table 4: Impact of coarse and fine vocabulary bag generation.**

| Train Set | Coarse | Fine | U-ARecall | K-mAP | UK-Mean |
|---|---|---|---|---|---|
| SODA | ✓ | | **77.3** | **65.9** | **71.6** |
| SODA | | ✓ | 59.5 | 64.3 | 61.9 |
| BDD100K | ✓ | | **68.3** | **60.5** | **64.4** |
| BDD100K | | ✓ | 59.7 | 58.0 | 58.8 |

Additionally, incorporating the VorCL scheme into our solution, results in a 1.1% improvement in K-mAP, proving its effectiveness in acquiring scenario-specific knowledge during fine-tuning.

**Sensitivity Analysis**. As illustrated in Fig. 9, we conducted experiments to analyze the sensitivity of hyper-parameters in our method. It is evident that our method is not sensitive to hyper-parameters.

**Discussion**. In Sec. 5.1.2, we define a general vocabulary bag with three coarse unknown classes for AD-oriented OWOD. An intuitive question is whether refining the class vocabularies in the vocabulary bag can improve performance. To investigate this issue, we

also design a more delicate method to generate a fine vocabulary bag, which contains texts of 59 finer classes in total, and compare its performance with our default vocabulary bag. As shown in Tab. 4, it is obvious that our default vocabulary bag is much better than the refined vocabulary bag. This indicates that having more vocabularies is not necessarily better. It increases the risk of more false positives, as the OVM may not be able to correctly recognize all classes in the vocabulary bag.

## 7 Conclusions

This paper address a core challenge that impacts the safety of AD, *i.e.*, detecting threatening open-world objects in AD. To address this challenge, we make three-level contributions: 1) On the level of evaluation, we mitigate the ambiguity of labeling unknown objects by restricting them to threatening objects, and prevent the misleading evaluation in original OWOD by introducing a new evaluation metric named U-ARecall. 2) On the level of overall OWOD paradigm, we alleviate the difficulty of distinguishing between unknown objects and false positives by exploiting the rich multimodal knowledge of OVMs, and adapt them to the AD scenario by fine-tuning. 3) On the level of specific solution, we identify two challenges when fine-tuning OVMs: catastrophic forgetting and scenario-specific knowledge acquisition. By addressing these challenges, our method achieves state-of-the-art performance for AD-oriented OWOD.

# 8   ACKNOWLEDGEMENT

This research was funded by the National Key Research and Development Program of China (No.2018YFB0204301) and the Natural Science Foundation of Hunan Province of China (No. 2022JJ30666).

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
