# OpenReview forum: "Sniffing Threatening Open-World Objects in Autonomous Driving by Open-Vocabulary Models"
_acmmm.org/ACMMM/2024/Conference — MM2024 Poster_

### Official Review · Reviewer_249B · 2024-05-14

**Rating:** 3
**Confidence:** 3

**Summary:**

The authors proposed a novel framework for detecting threatening open-world objects by introducing an open vocabulary model.  This framework addresses two core challenges of fine-tuning open vocabulary models (OVMs) through two innovative techniques: 1) the preservation of open-world generic knowledge using a dual-branch architecture;  and 2) the acquisition of scenario-specific knowledge via a visually-oriented contrastive learning scheme.  Extensive experiments and analyses on benchmark datasets demonstrate that the proposed methods achieve state-of-the-art performance.

**Strengths:**

1. The paper's motivation is highly novel and addresses a fundamental challenge that significantly impacts the safety of autonomous driving.
2. Numerous experiments have been conducted to demonstrate the efficacy of the method.
3. Open vocabulary model offers a novel perspective for sniffing threatening open-world objects.
4. The writing style and structure of the paper make it easy for readers to comprehend the methodology and significance of the study.

**Limitations:**

1. The confidence level of the car and person in Figure 8 is too low. A proficient object detection model designed for detecting hazards should consistently uphold a high level of accuracy across the annotated categories.
2. The paper fails to provide a reasonable explanation of how the model prevents misclassifying obstacles as background during the fine-tuning process in the training phase.
3. The paper lacks a reasonable explanation of how the model can differentiate obstacles from the background during the inference phase, and instead solely relies on frozen OVMs.
4. Whether the specific classification of obstacles depends on the existing knowledge in OVM？
5. The visualization results presented in the supplementary material appear to be the outcome of a direct test of the 59 finer classes using the existing OVM.
6. The experiment lacks unknown class mAP.


Further Comments

1. I hope the author takes the rebuttal seriously, and I will revise the review as appropriate based on the response provided in the rebuttal.
2. The paper need cite more papers related to multimedia.

**Suitability:**

3

---

### Official Review · Reviewer_7JWU · 2024-05-25

**Rating:** 4
**Confidence:** 3

**Summary:**

This is a paper that fully considers open-world object detection in autonomous driving scenarios. First of all, based on the characteristics of autonomous driving scenarios, the article proposes a new indicator u-arecall to alleviate the evaluation bias caused by ignoring false alarms. Afterwards, this paper provides rich prior information for the OWOD task by introducing an open vocabulary model and fine-tuning it. Finally, the paper optimizes the fine-tuning process and achieves good results.

**Strengths:**

(1) This paper will propose new indicators adapted to autonomous driving scenarios.
(2) This paper introduces efficient parameter migration to make LLM better adapt to the current scenario. Compared with overall fine-tuning, the method in this paper avoids both overfitting and large-scale parameter updates.

**Limitations:**

In the experimental part, Visual Prompt Tuning should be introduced to further illustrate the effectiveness of the method in this paper.

**Suitability:**

3

---

### Official Review · Reviewer_Jhu7 · 2024-06-02

**Rating:** 5
**Confidence:** 2

**Summary:**

This paper aims to address the challenge of detecting unknown, potentially threatening objects in autonomous driving (AD). Existing methods for open-world object detection (OWOD) often result in excessive false positives, which pose a safety risk for AD. To tackle this, the authors propose a new evaluation protocol focused on threatening objects specifically related to AD, introducing the U-ARecall metric to account for false positives more effectively. The paper also presents a novel OWOD paradigm based on fine-tuning foundational open-vocabulary models (OVMs). The proposed solution includes a dual-branch architecture to maintain open-world knowledge, a visual-oriented contrastive learning scheme to acquire scenario-specific knowledge, and a prediction fusion module to avoid manual post-processing. The authors demonstrate through extensive experiments that their method significantly outperforms existing OWOD methods in terms of both U-ARecall and K-mAP.

**Strengths:**

1. The introduction of the U-ARecall metric addresses the critical issue of false positives in OWOD, which is particularly important for the safety of AD.

2. The innovative use of a dual-branch architecture helps in retaining general open-world knowledge while fine-tuning for specific scenarios, mitigating the problem of catastrophic forgetting.

3. This technique enhances the model's ability to acquire scenario-specific knowledge, improving the detection of both known and unknown objects.

4. The authors conduct thorough experiments on different datasets, demonstrating the robustness and effectiveness of their approach across various domain gaps.

5. The proposed method significantly outperforms existing methods, achieving approximately three times the U-ARecall and a notable improvement in K-mAP.

**Limitations:**

1. The proposed dual-branch architecture and visual-oriented contrastive learning add complexity to the model, which may impact the practicality of deployment in real-world scenarios.

2. While the method performs well on selected datasets, further validation on a wider range of datasets and real-world conditions would strengthen the generalizability claims.

3. Although the sensitivity analysis shows robustness to hyperparameter changes, the method still relies on several hyperparameters, which might require fine-tuning for different applications.

**Suitability:**

2

---

### Meta-Review · Area_Chair_aMum · 2024-06-30

**Recommendation:** Accept (Poster)
**Confidence:** 5

**Metareview:**

This paper aims to address the challenge of detecting unknown, potentially threatening objects in autonomous driving. Its pros including the introduction of U-ARecall metric, innovative use of a dual-branch architecture, efficient parameter migration, thorough experiments, good results, and etc. The reviewers raised concerns regarding model complexity, results relying on several hyperparameters, etc. The authors well address these concerns in the rebuttal. Reviewer 249B gave borderline reject but had some vital misunderstandings. Given the strong recommendation from Reviewer Jhu7 and borderline accept recommendation from Reviewer 7JWU, the AC recommends this paper for acceptance.